# Iterative-Based Impact Force Identification on a Bridge Concrete Deck

**DOI:** 10.3390/s23229257

**Published:** 2023-11-18

**Authors:** Maria Rashidi, Shabnam Tashakori, Hamed Kalhori, Mohammad Bahmanpour, Bing Li

**Affiliations:** 1Centre for Infrastructure Engineering, Western Sydney University, Kingswood, NSW 2747, Australia; m.rashidi@westernsydney.edu.au; 2Department of Mechanical Engineering, Shiraz University of Technology, Shiraz 71557-13876, Iran; tashakori@sutech.ac.ir; 3Department of Mechanical Engineering, Faculty of Engineeinrg, Bu-Ali Sina University, Hamedan 65167-38695, Iran; 4School of Mechanical and Mechatronic Engineering, University of Technology Sydney, Sydney, NSW 2007, Australia; 5Department of Mechanical Engineering, Shiraz University, Shiraz 1585-71345, Iran; mohammadbahmanpour@hafez.shirazu.ac.ir; 6School of Aeronautics, Northwestern Polytechnical University, Xi’an 710072, China

**Keywords:** impact force identification, impact localization, structural health monitoring, bridge concrete deck, Landweber method, iterative regularization

## Abstract

Steel-reinforced concrete decks are prominently utilized in various civil structures such as bridges and railways, where they are susceptible to unforeseen impact forces during their operational lifespan. The precise identification of the impact events holds a pivotal role in the robust health monitoring of these structures. However, direct measurement is not usually possible due to structural limitations that restrict arbitrary sensor placement. To address this challenge, inverse identification emerges as a plausible solution, albeit afflicted by the issue of ill-posedness. In tackling such ill-conditioned challenges, the iterative regularization technique known as the Landweber method proves valuable. This technique leads to a more reliable and accurate solution compared with traditional direct regularization methods and it is, additionally, more suitable for large-scale problems due to the alleviated computation burden. This paper employs the Landweber method to perform a comprehensive impact force identification encompassing impact localization and impact time–history reconstruction. The incorporation of a low-pass filter within the Landweber-based identification procedure is proposed to augment the reconstruction process. Moreover, a standardized reconstruction error metric is presented, offering a more effective means of accuracy assessment. A detailed discussion on sensor placement and the optimal number of regularization iterations is presented. To automatedly localize the impact force, a Gaussian profile is proposed, against which reconstructed impact forces are compared. The efficacy of the proposed techniques is illustrated by utilizing the experimental data acquired from a bridge concrete deck reinforced with a steel beam.

## 1. Introduction

Structural Health Monitoring (SHM) has gained increasing attention over the past decades due to its applicability to various infrastructures such as bridges [1,2], railways [3,4], skyscrapers [5,6], etc. One of the applications of SHM is the prediction of failure as a result of accidental impact forces imposed on structures. Direct measurement of these impact events, however, is not always possible since there are limitations on the number of sensors mounted on a structure and their potential places. Hence, using dynamic structural responses to inversely reconstruct the impact forces is a promising alternative approach.

Impact force reconstruction methods are usually ill-posed, which results in sensitivity to measurement noises. To deal with the ill-posedness, several regularization techniques are exploited in the literature, such as the Tikhonov method [7,8], Truncated Singular Value Decomposition (TSVD) [9], and Bayesian regularization [10,11]. Finding an optimum regularization parameter plays an important role in the regularization performance, for which various techniques like Generalized Cross Value (GCV) [12], L-curve [13], l_1_ and l_2_ norm [14,15], etc., are proposed. Alternatively, there is another class of regularization technique known as the Iterative Regularization method [16,17], which iteratively finds the regularization parameter.

Many iterative regularization methods are presented in the literature, such as the Landweber method [18], the Kaczmarz method [19,20], and the Krylov subspace methods [21], including Conjugate Gradients [22], Least Square QR (LSQR) [23], and LSMR [24]. The main superiorities of iterative regularization methods are that, firstly, they avoid extensive regularization parameter selection and hence are preferable for large-scale problems [21], and, secondly, their solution is in general more reliable and accurate. 

Due to their faster convergence, Krylov subspace methods have recently gained more attention compared with the Landweber and Kaczmarz techniques [25]. Nevertheless, due to more semi-convergence behavior, the Krylov subspace method is prone to poor solution accuracy without appropriate stopping criteria [26]. On the other hand, the Landweber method is still superior in terms of simplicity and stability, which makes it more suitable in some real-world applications [27,28]. An adaptive Landweber method [29] has also been proposed in order to accelerate the convergence of this technique. However, its performance is not improved in case of the lack of a proper preset parameter. In [30], the implicit Landweber method has been employed to reconstruct dynamic force exerted on a thin-walled square. Therein, it is concluded that proper initial parameters and convergence criteria should be chosen in order to arrive at an optimal force reconstruction. As the convergence criteria highly depend on the noise level, the results should be processed in advance to obtain the level of noise in the response.

In this paper, the impact force reconstruction is performed on a bridge concrete deck experimental setup that is reinforced with a steel beam. This experimental setup models the steel-reinforced concrete decks that are used in bridges and railways. Herein, the Landweber method is exploited to relax the ill-conditioning of the identification problem, and the solution is found iteratively. The efficacy of the impact reconstruction and regularization methods are illustrated based on the experimental data. Furthermore, a discussion on the selection of (i) the sensor location and (ii) the number of iterations is presented. The impact localization is also investigated based on the proposed reconstruction strategy.

The main contributions of this paper are as follows: It is shown that introducing a low-pass filter to the Landweber-based impact reconstruction can improve the reconstruction precision. The idea behind this introduction relies on the fact that ill-posedness of the reconstruction problem leads to sensitivity to measurement noises. Therefore, as will be discussed, filtering the high-frequency contents in the response signal can benefit the regularization problem and hence the reconstruction precision.A standardized accuracy error metric is utilized that improves the evaluation of the reconstruction validity. This metric regards both the correlation and peak error and hence can lead to more accurate evaluation than some other error metrics exploited in the literature.The impact localization can be performed in an automated manner by using a proposed Gaussian profile. This idea relies on the fact that the overall shape of an impact force can be considered similar to a Gaussian profile. Even in the presence of damage, some local fluctuations will be added to this global impact profile [31,32]. We believe that the proposed Gaussian profile matches the global shape and impact force more precisely compared with the half-sine signal employed in the literature [33,34].

The paper is organized as follows: Section 2 presents the overall idea of impact force localization as well as the reconstruction of the impact time–history. Therein, the Landweber method is also introduced. In Section 3, the experimental setup utilized in this paper is presented. Section 4 shows the simulation results, including the discussion on the results. This paper is concluded in Section 5.

## 2. Identification of the Impact Force

The complete identification of an impact force is pursued in two phases, as follows:Phase 1: localizing the impact force;Phase 2: reconstruction of the impact force time–history.

Most of the impact force reconstruction techniques need the impact location a priori. Therefore, in the following, we first present the localization approach utilized in this paper (Phase 1) and next the reconstruction strategy (Phase 2).

### 2.1. Impact Force Location

Given the total number of *n* impact locations and *p* response measurement points, the global transfer matrix of a system is represented as follows [33]:(1)$$T=\left[\begin{array}{ccc} T_{11} & \cdots & T_{1n}\\ \vdots & \ddots & \vdots \\ T_{p1} & \cdots & T_{pn} \end{array}\right],$$
which mathematically characterizes the relationship between each pair of impact force $f_{j}$ applied at location *j* and the vibration response $r_{i}$ acquired at position *i*, i.e.,
(2)$$r_{i}=T_{ij}f_{j},i=1,\ldots ,p\;\mathrm{and}\;j=1,\ldots ,n.$$

Suppose that the response of an unknown impact force is measured at position *i*. To localize the impact force, impact force reconstruction is performed *n* times for each of the potential impact location at j=1,…,n. Practically, one of the reconstructed impact forces will have a similar profile as an actual impact force, which determines the true impact location [31,33]. Note that, ideally, other reconstructed impact forces are expected to have near-zero magnitude and/or a non-sinusoidal profile [31,33].

To perform the above impact localization strategy, the transfer matrix ***T*** needs to be created in advance. This is usually carried out by identifying ***T****_ij_* using a reference impact force ***f****_j_* with known magnitude and location, as well as its resulting vibration response ***r****_i_*. Note that both transfer function identification and impact force localization methods discussed above are ill-conditioned, i.e., sensitive to measurement noises and perturbations. As mentioned earlier, in this paper, both problems are regularized in order to arrive at a stable solution, which will be presented in Section 2.3.

### 2.2. Impact Force Time–History

The impact force–response relationship in Equation (2) can be rewritten in an inclusive manner, as follows [33,34]:(3)R=TF,
where $R^{mp\times 1}$ is a vector containing all acquired responses by available sensors, $F^{mn\times 1}$ is the corresponding impact force/forces vector, and the discretized transfer matrix $T^{mp\times mn}$ as defined in Equation (1), with m number of collected samples. Equation (3) is first solved for the transfer matrix ***T*** by using as reference the known impact forces and their resulting responses. Then, with the transfer matrix ***T*** known, Equation (3) is solved for an unknown impact force based on its corresponding collected responses. It is worth mentioning again that the reconstruction problem discussed is ill-conditioned and hence needs to be regularized. In the following, the regularization approach utilized in this paper is introduced.

### 2.3. Landweber Regularization

Consider a linear ill-posed problem with the following general form:(4)$$\tilde{B}=Ax,$$
where the exact vector ***B*** is perturbed, e.g., by measurement errors, shown by $\tilde{B}$. The solution to this problem will be unstable, as the errors will be amplified, and generally, it cannot be recovered. Therefore, regularization methods should be employed to relax the ill-conditioning, which normally seeks an approximated solution, i.e., the underlying problem is as follows:(5)$$\tilde{B}\approx Ax,$$

As mentioned in Section 1, the regularization approaches in the literature can be categorized into two groups: **Direct approach**, including Tikhonov and TSVD methods;**Iterative approach**, such as Landweber and Krylov subspace methods.

In the direct approach, matrix ***A*** is decomposed by the Singular Value Decomposition (SVD) technique and the problematic modes (e.g., corrupted with noise) are excluded. The performance of these methods relies on the selection of the regularization parameter. On the other hand, methods based on iteration do not need the SVD, which is computationally expensive when the dimension of the matrix A is large. In this approach, the number of iterations is responsible for the solution convergence and accuracy, similar to the regularization parameter in the direct approach.

Iterative regularization methods consent to a phenomenon called semi-convergence, which is referred to the case of terminating the regularization problem before the asymptotic convergence. As shown in Figure 1, in the iteration number *k_opt_*, the corresponding solution $x^{\left[k_{opt}\right]}\;$ is the closest value to the exact solution $x^{exact}$. While, if the iteration continues to the solution $x=A^{-1}\tilde{B}$, which is referred to the asymptotic convergence, the solution will be less accurate. This is what occurs in regularization methods with fixed regularization parameters [7,8,9,10,11,12,13,14,15], which leads to a naive solution and, in some cases, may even diverge.

The iterative regularization algorithms seek a solution $x^{\left[k\right]}$ where
(6)$$x^{\left[k\right]}=argmin_{x}Ax-\tilde{B}.$$

Equation (6) means the solution *x* that results in the minimum value of $Ax-\tilde{B}$ [35]. Indeed, *argmin_x_* is defined as the input to a function f(x) that yields the minimum value of that function, i.e.,
(7)$$argmin_{x}=\left\{x\;\;s.t.\;f\left(x\right)=\underset{X}{\min}f\left(X\right)\right\},$$
where *X* includes any possible input *x*. In the basic form of the Landweber regularization method, the solution of Equation (6) is found as follows [35]:(8)$$x^{\left[k+1\right]}=x^{\left[k\right]}+\omega A^{T}\left(\tilde{B}-Ax^{\left[k\right]}\right),$$
where the real number ω satisfies the following condition:(9)$$0<\omega <2A^{T}A_{}^{-1}.$$

## 3. Experimental Setup

Steel-reinforced concrete decks have been extensively used in many civil structures such as bridges and railways, which are subjected to unknown accidental impact forces during their service life. In this regard, an experimental setup composed of a concrete deck reinforced with a steel beam is utilized in this paper to show the efficacy of the impact force identification introduced in Section 2.

The setup, shown in Figure 2, with its cross-sectional view shown in Figure 3, is designed and fabricated in a manner that simulates the decks of the Sydney Harbour Bridge in Australia [36]. The deck is 2 m in length and 1 m in width. The steel I-beam (200 UB 18) is 1900 mm in length. Several initiatives were proposed to facilitate the structural health monitoring of the Sydney Harbour Bridge in Sydney, Australia. As part of these endeavors, the bridge was instrumented with an extensive array of sensors. Additionally, a simulated deck, mirroring the characteristics of the bridge’s actual decks, was constructed at the University of Technology Sydney. This controlled environment enabled the execution of a wide range of tests that would otherwise be challenging to conduct in the field.

The deck was clamped at one end and gridded so that the impact location could be more easily identified. The potential impact locations were numbered from L_1_ to L_7_, as shown in Figure 2, and were assumed to be known a priori. These points were selected totally arbitrarily. In every impact location, multiple impact forces were exerted with a PCB impact hammer (model 086D20). Note that, to verify different load cases, a series of experiments was systematically conducted, aiming at achieving a wide spectrum of impact forces characterized by varying amplitudes and durations contingent upon the specific material properties of the hammer tips. The objective was to ensure the inclusion of a comprehensive range of impact force scenarios, thus enhancing the diversity of the experimental dataset.

The acceleration responses of impact forces were acquired by 10 accelerometers mounted at specific measurement points, shown by S_1_ to S_10_ in Figure 2. These accelerometers were attached to the deck using a strong magnet and metallic plate. A threshold of 200 N was utilized for triggering the signal. Given the impact hammer’s sensitivity of 2.27 mv/N, an applied impact force of 200 N yielded an output voltage of 0.454 volts. Lowering the trigger threshold led to inadvertent signal triggering within the acquisition system, attributed to ambient noise and cable-induced motion.

## 4. Results and Discussion

### 4.1. Discussion on Impact Force Reconstruction

Reconstruction of impact forces applied at locations L_1_ to L_7_ are shown in Figure 4, Figure 5, Figure 6, Figure 7, Figure 8, Figure 9 and Figure 10, respectively. In these figures, the impact force is reconstructed 10 times by using each individual accelerometer S_1_ to S_10_. Here, the Landweber method is employed in order to regularize the identification problem with 2500 iterations. Moreover, aiming at a more accurate reconstruction, the initially reconstructed impact force is filtered by a low-pass filter (with 0.01 passband frequency, 0.35 stop band frequency, 0.5 passband ripple, and 65 stop band attenuation, designed based on the properties, i.e., the frequency of high-frequency contents, of the response signals). 

Let us take a closer look. As can be seen in Figure 4, the impact force exerted at L_1_ is best reconstructed using accelerometer S_4_, while S_6_ completely fails to reconstruct this impact force. The same evaluation can be performed for other figures (impact forces exerted at other locations), as well. For example, for the impact location L_2_, accelerometer S_9_ is the best choice to be used for impact reconstruction, i.e., results in more accurate reconstruction, whereas accelerometers S_1_, S_3_, and S_8_ are the worst choices. Roughly speaking, here, it can be concluded that it is generally more difficult to reconstruct the impact exerted at L_2_ compared with L_1_. The similar sketchy evaluation of other impact locations is not presented here for the sake of brevity but will be summarized in the following paragraphs.

As can be seen in the above figures, the impact force cannot necessarily be reconstructed using any arbitrary accelerometers. For example, S_7_ can identify the impact forces applied at all potential locations except L_4_ with less than 10 percent reconstruction error, while, for the same level of accuracy, S_3_ can only be utilized to identify the impact forces applied at L_6_ and L_7_. This can be influenced by many factors, such as (i) the quality of sensor attachment, (ii) the complexity of the dynamic characteristics of the reinforced concrete deck in different directions that affects the quality of the wave propagation, (iii) the presence of disturbances, etc. However, for a particular configuration, this procedure can be elucidated through the implementation of pre-established tests similar to those conducted within this study. Consequently, one can deduce the optimal selection of measurement data to effectively reconstruct an unknown impact force at a designated location.

To evaluate the performance of each sensor in the reconstruction precision at each impact location, Table 1 is presented. In this table, the cells showing less than 5 percent reconstruction error are colored green, the cells with between 5 to 10 percent error are colored yellow, and the red cells are the ones with more than 30 percent error. As shown, the impact forces applied at L_7_ and L_6_ have been reconstructed satisfactorily with most accelerometers, while the impact force at L_4_ is the most difficult to be identified. It is also concluded that, generally, accelerometers S_7_ and S_4_ give the most reliable result for every impact location compared with other accelerometers. On the other hand, using accelerometers S_1_, S_3_, S_8,_ and S_10_ leads to less precise impact reconstruction, so they can be excluded. To sum up, using this table, one can choose which measurement to utilize for reconstruction once the impact location is known. It is worth mentioning that although the number of iterations influences the reconstruction error, it does not invalidate the above conclusion, as will be discussed later.

The reconstruction error presented above each subplot in Figure 4, Figure 5, Figure 6, Figure 7, Figure 8, Figure 9 and Figure 10, and also in Table 1, is defined as follows [33]:(10)$$\%e=\sqrt{{\left(1-cv\right)}^{2}+e_{p}^{2}}\times 100,$$
where *e_p_* is the peak error indicating the difference between the peak values of the actual and reconstructed impact forces, and 1−cv is the correlation error with cv the correlation value indicating the similarity between the actual and reconstructed signals. In other words, the impact force reconstruction is more accurate when the reconstruction error is closer to zero percent. This is achieved when the correlation value and the peak error are, respectively, closer to one and zero. Note that neither of the quantities cv or *e_p_* can solely show the effectiveness of the reconstruction, as discussed in [34]. For instance, the profile of the reconstructed impact force may be very similar to the profile of the actual impact force (i.e., cv is very close to one), while their peak values have a considerable difference. This case is not considered an accurate reconstruction. Similarly, the reconstructed and actual impact forces may have almost the same peak values (i.e., *e_p_* close to zero), while the rest of their profiles are not similar. This is also not considered a precise reconstruction.

### 4.2. Discussion on Landweber Regularization

This section aims at, firstly, enlightening the effect of the number of iterations in the Landweber method on the reconstruction accuracy and, secondly, revealing the reason we introduced such a low-pass filter to the Landweber-based reconstruction procedure. 

Figure 11 and Figure 12, respectively, show the correlation error and the peak error of the impact reconstruction employing the Landweber regularization method with different numbers of iterations, ranging from 5 to 4905. In these figures, the blue dashed lines correspond to the initial reconstruction (that employs the Landweber regularization without any filtering). The black line corresponds to the filtered reconstruction, that is, the initially Landweber-based reconstructed impact force is filtered by a low-pass filter. As illustrated, applying a low-pass filter to the reconstruction procedure enhances the accuracy both in terms of correlation and peak conformity. The reconstruction in these figures is performed by utilizing the accelerometer S_1_. However, the above conclusion is valid not only for all potential impact locations, as can be seen in Figure 11 and Figure 12, but also for all accelerometers. It should be noted that the results of other measurements are not presented here for the sake of brevity.

The effect of the number of iterations on the reconstruction error is investigated as follows. Figure 13, Figure 14, Figure 15, Figure 16, Figure 17, Figure 18 and Figure 19 demonstrate the reconstruction error using the Landweber regularization in combination with a low-pass filter for impact locations L_1_ to L_7_, respectively. As can be seen, the number of iterations has a considerable influence on the reconstruction error. However, to make a fair comparison between different sensor locations and impact locations, a certain number of iterations should be used in all cases. Therefore, since, in the majority of the instances, substantial changes occur almost before 2500 iterations, this number of iterations is used to obtain Figure 4, Figure 5, Figure 6, Figure 7, Figure 8, Figure 9 and Figure 10. It is noteworthy to mention that, as can be seen, smaller values for the number of iterations can also lead to satisfactory results. In other words, it is not claimed that the number 2500 is an optimal value, it is just selected to make other comparisons consistent.

It is observed in Figure 13, Figure 14, Figure 15, Figure 16, Figure 17, Figure 18 and Figure 19 that the reconstruction error is minimum at a certain number of iterations for each case. Therefore, two scenarios can be pursued to choose the number of iterations, as follows:**Scenario 1:** Pre-made tests can be exploited to obtain the optimal value of the number of iterations for each combination of the impact location and measurement point, as performed in this section. Consequently, the most precise reconstruction can be achieved, which can benefit applications that rely on high reconstruction accuracy.**Scenario 2:** A specific iteration number can be employed for all possible combinations of the impact location and measurement point, as performed in Section 4.1. Roughly speaking, this can be conducted especially when there is a relative enough number of sensors available. More precisely, as presented in Table 1, for each impact location there exists at least one sensor that yields the reconstruction error of less than 10%. Although it might not be the most accurate reconstruction possible, this level of accuracy is acceptable in many applications.

The least reconstruction error that can be obtained for the number of iterations less than 4905 is shown in Table 2 for all potential locations using all available accelerometers. The cells colored in green demonstrate less than 5 percent error, those colored in yellow correspond to errors between 5 to 10 percent, and errors greater than 30 percent are colored in red. Obviously, the errors are less than presented in Table 1, for which a constant number of iterations is used (Scenario 2), since, in this table, an optimal value is used in each case (Scenario 1). The conclusions made previously are still valid. In other words, it is still concludable that impacts exerted at L_7_ and L_6_ are reconstructed with high accuracy, unlike L_4_, which leads to the least precision reconstruction. It is still valid that accelerometers S_7_ and S_4_ totally give reliable results, while S_1_, S_8_, and S_10_ are less trustworthy. However, the results correspond to impact location L_2_, and the accelerometer S_3_ is considerably improved compared with Table 1 (Scenario 2). In conclusion, although pursuing Scenario 1 can lead to more precise reconstruction results, Scenario 2 can be sufficiently reliable and utilized in applications that do not rely on very high accuracy and where there are relatively enough sensors mounted on the system.

### 4.3. Discussion on Impact Localization

Based on the discussion in the previous section, and for the sake of brevity, the impact localization is performed based on the data acquired by accelerometer S_7_, which has been shown to be one of the most reliable measurements in solving the reconstruction problem for this setup (see Table 2).

Figure 20 illustrates that the impact localization approach presented in this paper can effectively localize the impact at all potential locations. In other words, in each subplot, the impact reconstruction is performed for all impact locations separately, using the measured data from accelerometer S_7_, while only one of them is the true impact location. To find the actual impact location, the reconstructed forces are compared to a Gaussian profile as follows:(11)$$F_{G}=a_{1}\exp\left(-{\left(\frac{x-b_{1}}{c_{1}}\right)}^{2}\right)$$
which has a similar profile as an actual impact force, with parameters *a*_1_ = 1100, *b*_1_ = 0.0515, and *c*_1_ = 0.0011, chosen arbitrarily. The errors shown in Figure 20 are the correlation errors. Note that the reconstructed force with a lower correlation error is more similar in shape to the Gaussian profile presented in Equation (11). Consequently, the location corresponds to the reconstructed force with a smaller correlation error is the true impact location. In Figure 20, the dashed blue line shows the Gaussian profile, the black solid line illustrates the reconstructed impact force at the true location, and other grey lines relate to the reconstructed impact forces at other locations. As shown, the reconstructed impact force at the true location has the smallest correlation error, except in one case. As can be seen, when the true impact location is L_4_, the strategy fails, since the reconstructed impact at L_3_ corresponds to a lower correlation error, which is shown by a red dotted line. This inaccuracy is due to the lower reconstruction accuracy at L_4_ by using S_7_, as shown in Table 2. As demonstrated in this table, S_7_ leads to a less precise identification for forces exerted at L_4_. Similarly, it can be concluded that, to best localize the impact forces at L_4_, the accelerometer S_4_ should be utilized. This is illustratively shown in Figure 21, where the impact force exerted at location L_4_ is effectively localized based on its minimum correlation error compared to the Gaussian profile.

### 4.4. Discussion on Sensor Placement

As demonstrated in Table 1 and Table 2, on the one hand, impact forces at L_7_, L_6_, and L_5_ have been reconstructed with more accuracy compared with other impact locations. On the other hand, it can be seen that S_7_ and S_9_ lead to higher reconstruction precision than other sensors. Referring to Figure 2, the aforementioned impact locations are the closest locations to the most effective sensors, S_7_ and S_9_. It can be hence concluded that the distance between the impact location and the measurement point plays an important role in the reconstruction accuracy. This may be due to the fact that the wave propagated as a result of the impact is less damped and affected by disturbances due to the relatively shorter distance it has traveled. The reason why S_7_ and S_9_ yield better reconstruction might be simply due to better sensor attachments or due to their placement at the boundary. Intuitively, these sensors are less subjected to wave reflection. In other words, the signal that other sensors receive is a combination of the original wave propagated and its corresponding reflective waves returned from boundaries, while, at a boundary, the wave acquired is more unadulterated. This rough conclusion would be stronger if S_10_ also gave precise results.

The impacts exerted at L_1_, L_3_, and L_4_ have been reconstructed more poorly compared with other impact locations. Similar to the above discussion, this could be because these locations are comparatively farther to sensors than other potential impact locations.

To sum up, in order to improve the impact reconstruction accuracy, the effect of (i) signal noise, (ii) disturbance, (iii) damping, and (iv) wave reflection should be taken into consideration and eliminated as much as possible. In this paper, it has been shown that filtering the measurement signals to reduce high-frequency contents corresponding to noise can be effective. The authors also predict that adding some vibration absorbers at the boundaries to stop reflecting waves can have a significant contribution to improving reconstruction accuracy.

### 4.5. Discussion on Real-World Applicability

The method presented in this paper is model-based. Therefore, similar to other model-based methods, it has a shortcoming that the impact identification efficacy relies on establishing accurate transfer matrices. These transfer matrices are created using reference known impact forces and their resulting responses. On the one hand, the reference impact load cannot be too large to avoid self-imposed damage. On the other hand, impact loads of interest in civil constructions are usually large in magnitude and produce large damages, significant inelastic response, and strain rate effects. Although with a less accurate model the reconstruction will be less reliable, this does not invalidate the present approach, as it has been previously shown that even if the transfer function is obtained by a lower-magnitude transfer function, the reconstruction can still be satisfactorily performed [33].

## 5. Conclusions

An impact force identification approach has been presented based on the iterative Landweber regularization method. To enhance the performance of this regularization technique, it has been proposed to add a low-pass filter to the reconstruction procedure. Moreover, a reconstruction error, rather than an, e.g., simple correlation error, is utilized to evaluate the overall efficacy of the proposed strategy. The proficiency of the identification method is shown by utilizing experimental data acquired from a steel-reinforced concrete deck setup. The effect of sensor placement and number of regularization iterations has been investigated on the reconstruction accuracy. It has been concluded that for impact identification at a particular location, particular sensors can lead to more precise results that can be discovered by pre-made tests similar to what has been performed in this paper. The impression of the number of Landweber iterations on the reconstruction accuracy has been also discussed. For a more careful identification, the same analysis can be performed in advance to identify the optimal number of iterations for a specific impact and sensor location. Otherwise, in applications that do not rely on very high reconstruction precision, an average number of iterations can be utilized. The proposed reconstruction strategy can be employed to localize an impact event. This claim has been validated illustratively based on the experimental data. To automatedly identify the true impact location, a Gaussian profile has been proposed, in which the reconstructed impact forces at all potential locations are compared. The reconstructed impact force with a minimum correlation error associated with the Gaussian profile reveals the impact location.

## Figures and Tables

**Figure 1 sensors-23-09257-f001:**
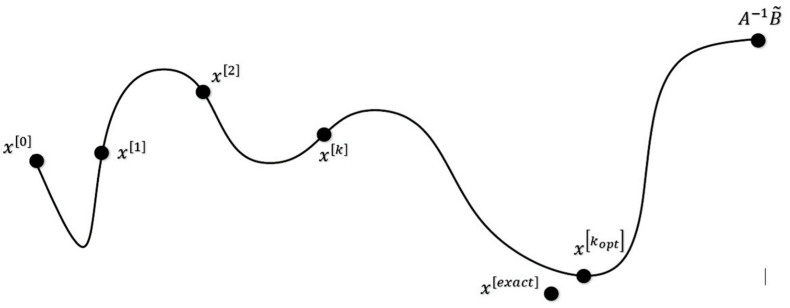
Error schematic showing asymptotic and semi-convergence phenomena [35].

**Figure 2 sensors-23-09257-f002:**
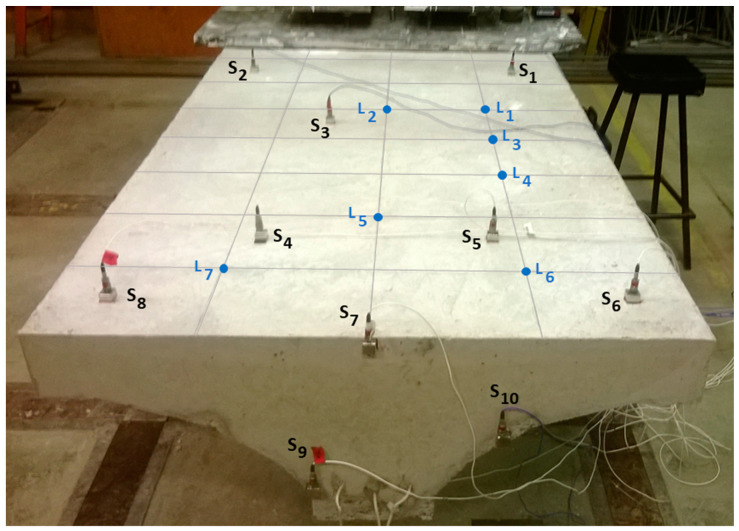
Experimental setup: a concrete deck reinforced with a steel beam with accelerometers placed at positions S_i_ (i = 1, …, 10), and the potential impact locations labeled as L_j_ (j = 1, …, 7).

**Figure 3 sensors-23-09257-f003:**
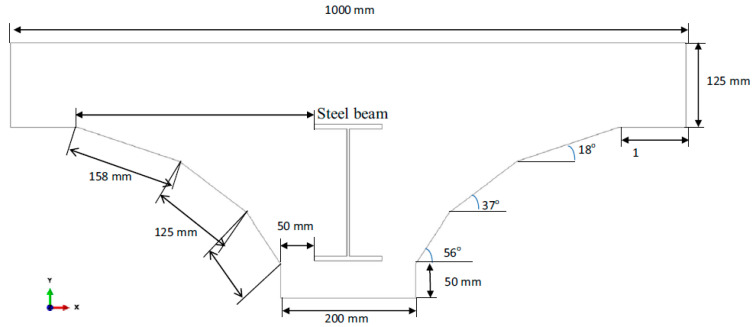
Cross-sectional view of the concrete deck setup.

**Figure 4 sensors-23-09257-f004:**
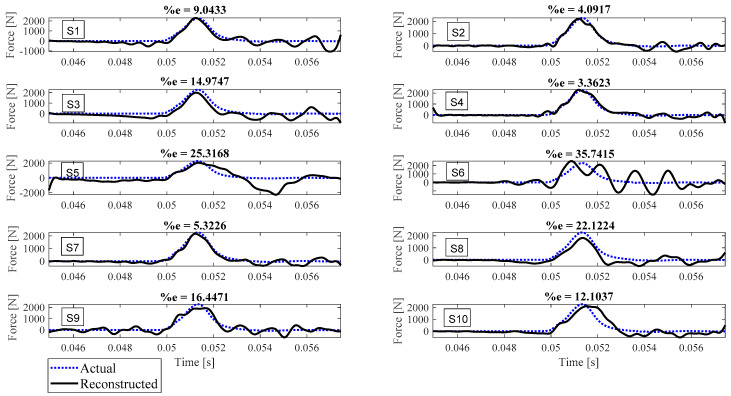
Reconstruction of the impact force applied at L_1_, using accelerometers placed at S_1_ to S_10_.

**Figure 5 sensors-23-09257-f005:**
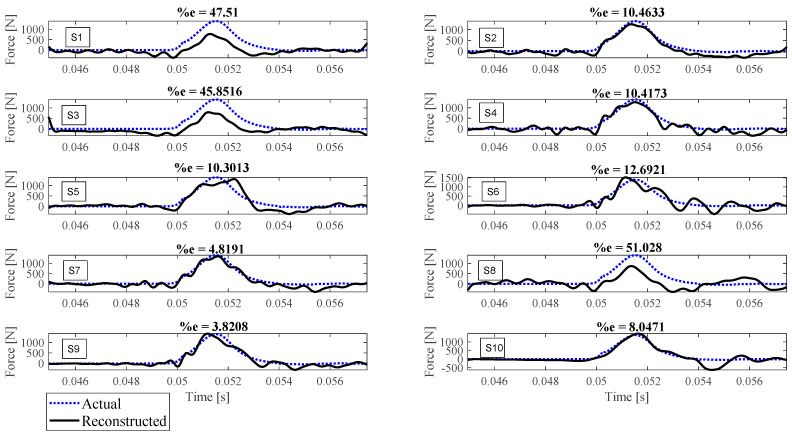
Reconstruction of the impact force applied at L_2_, using accelerometers placed at S_1_ to S_10_.

**Figure 6 sensors-23-09257-f006:**
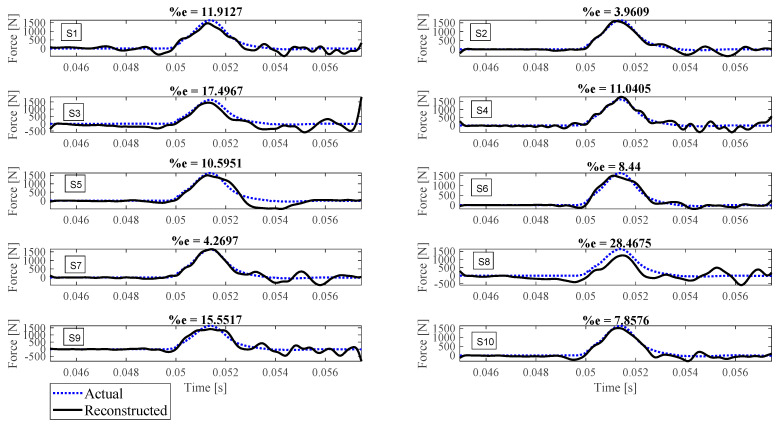
Reconstruction of the impact force applied at L_3_, using accelerometers placed at S_1_ to S_10_.

**Figure 7 sensors-23-09257-f007:**
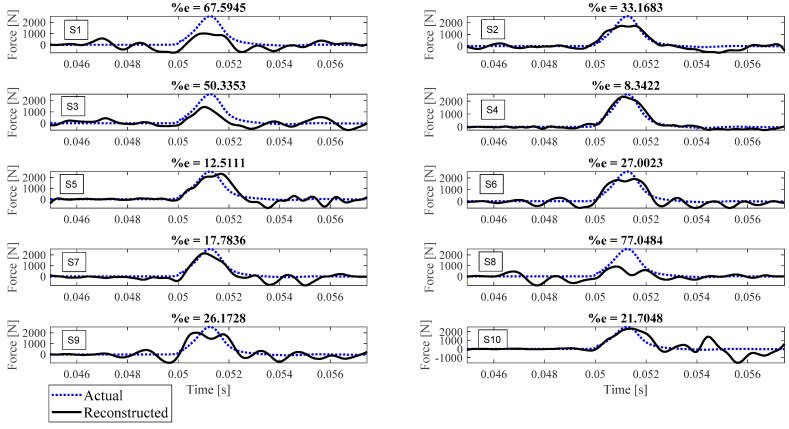
Reconstruction of the impact force applied at L_4_, using accelerometers placed at S_1_ to S_10_.

**Figure 8 sensors-23-09257-f008:**
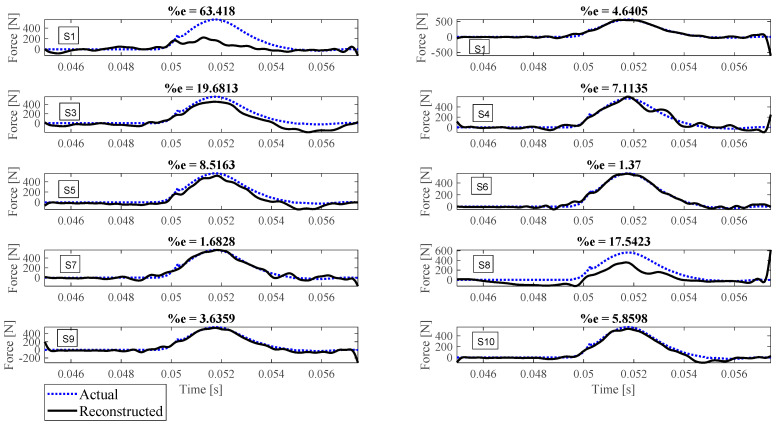
Reconstruction of the impact force applied at L_5_, using accelerometers placed at S_1_ to S_10_.

**Figure 9 sensors-23-09257-f009:**
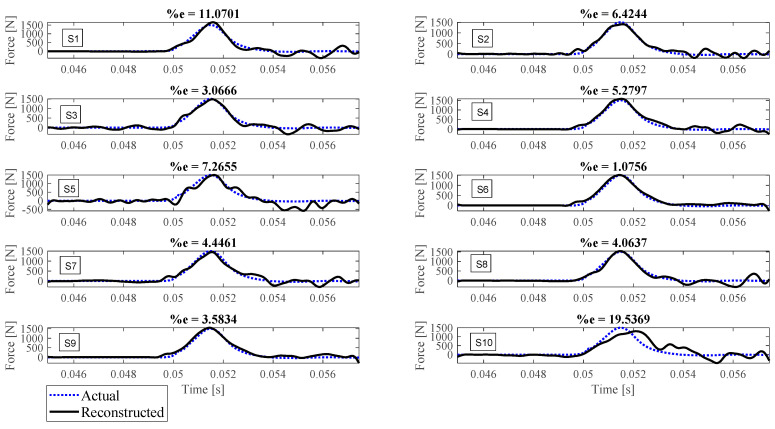
Reconstruction of the impact force applied at L_6_, using accelerometers placed at S_1_ to S_10_.

**Figure 10 sensors-23-09257-f010:**
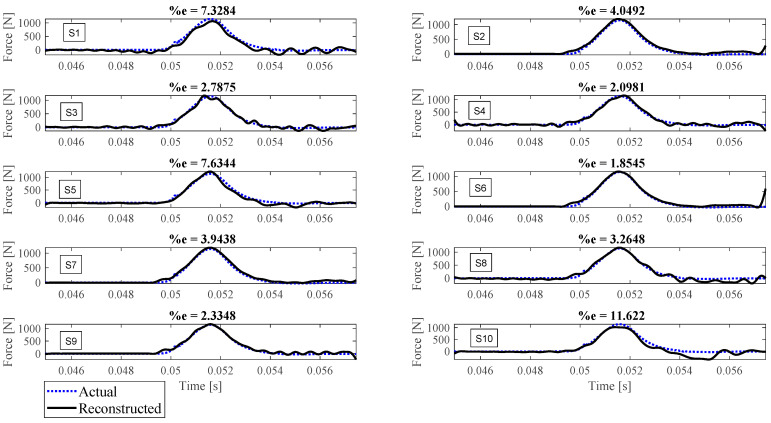
Reconstruction of the impact force applied at L_7_, using accelerometers placed at S_1_ to S_10_.

**Figure 11 sensors-23-09257-f011:**
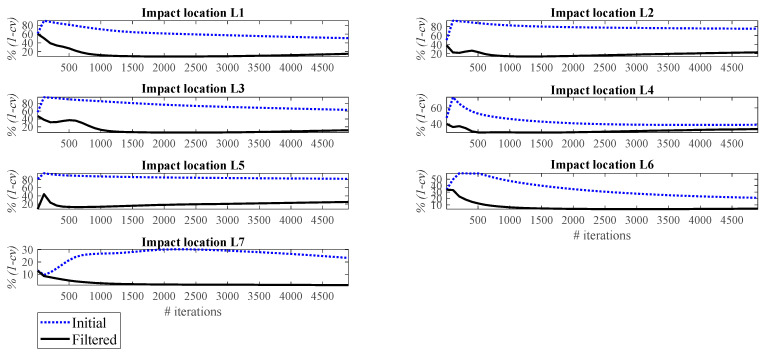
Correlation error for initial and filtered reconstruction at different locations using accelerometer S_1_.

**Figure 12 sensors-23-09257-f012:**
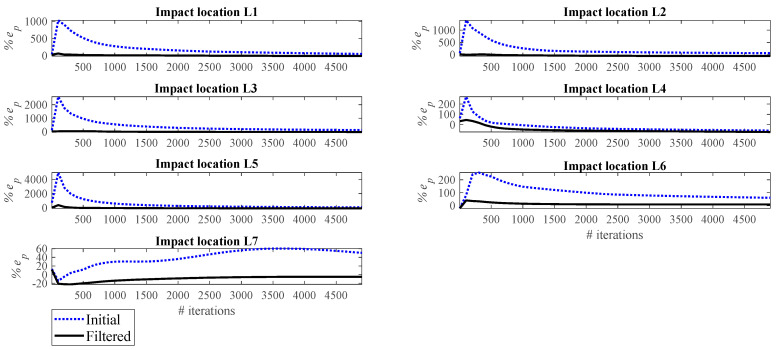
Peak error for initial and filtered reconstruction at different locations using accelerometer S_1_.

**Figure 13 sensors-23-09257-f013:**
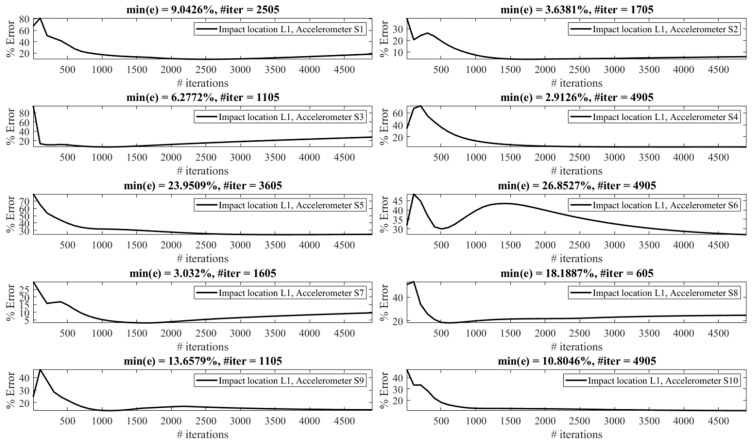
Reconstruction error at impact location L_1_ for different number of iterations in Landweber regularization, using accelerometers placed at S_1_ to S_10_.

**Figure 14 sensors-23-09257-f014:**
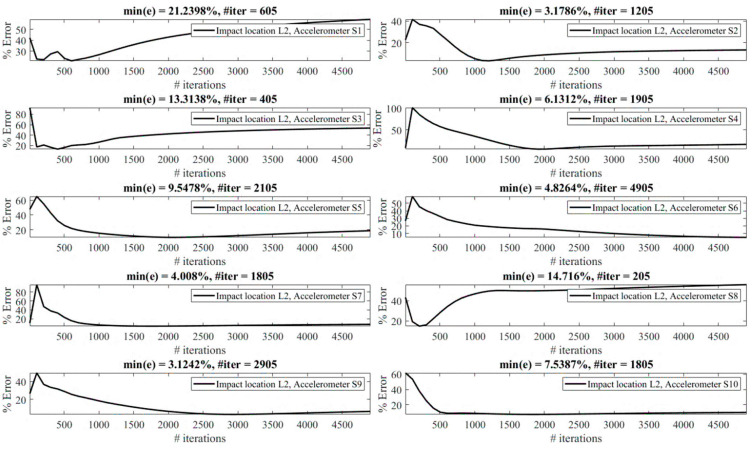
Reconstruction error at impact location L_2_ for different number of iterations in Landweber regularization, using accelerometers placed at S_1_ to S_10_.

**Figure 15 sensors-23-09257-f015:**
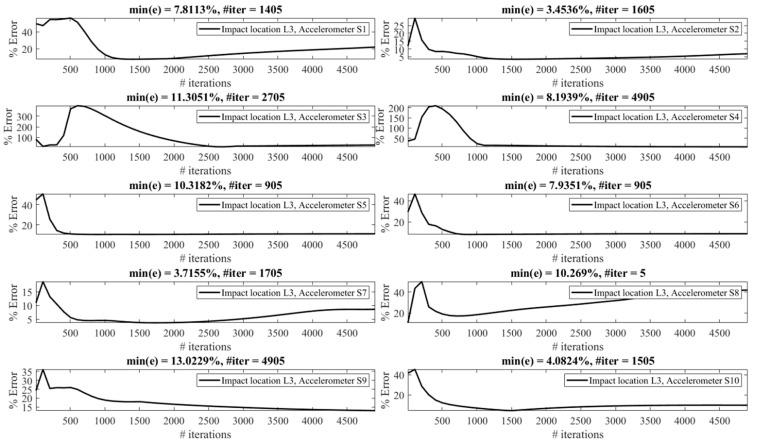
Reconstruction error at impact location L_3_ for different number of iterations in Landweber regularization, using accelerometers placed at S_1_ to S_10_.

**Figure 16 sensors-23-09257-f016:**
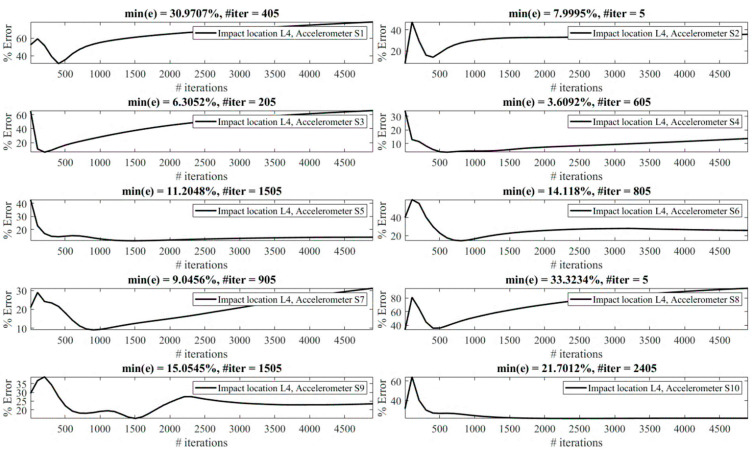
Reconstruction error at impact location L_4_ for different number of iterations in Landweber regularization, using accelerometers placed at S_1_ to S_10_.

**Figure 17 sensors-23-09257-f017:**
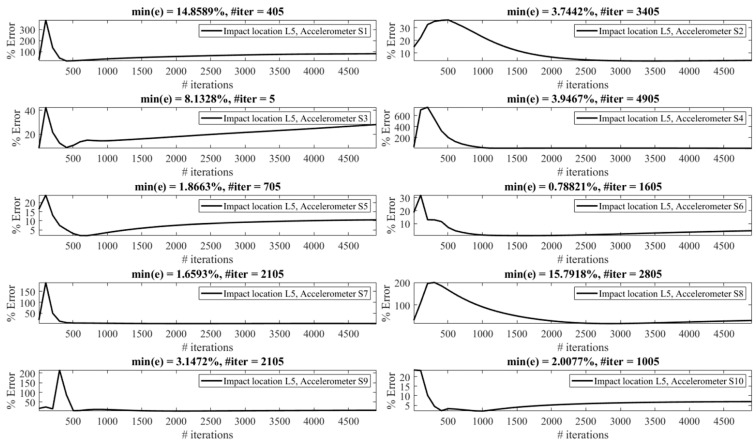
Reconstruction error at impact location L_5_ for different number of iterations in Landweber regularization, using accelerometers placed at S_1_ to S_10_.

**Figure 18 sensors-23-09257-f018:**
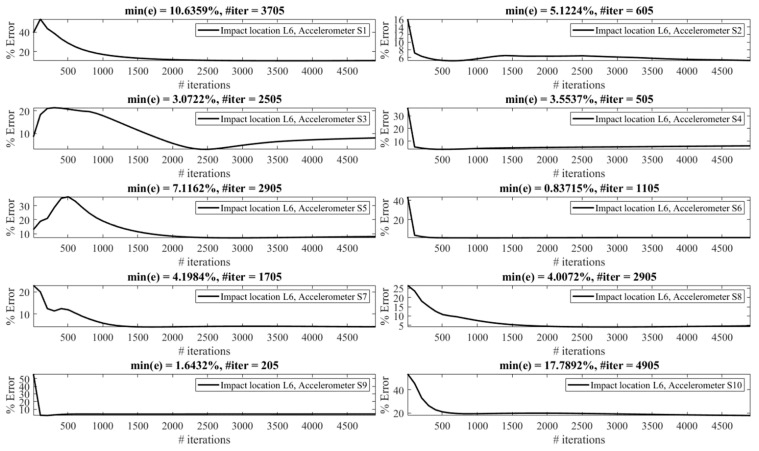
Reconstruction error at impact location L_6_ for different number of iterations in Landweber regularization, using accelerometers placed at S_1_ to S_10_.

**Figure 19 sensors-23-09257-f019:**
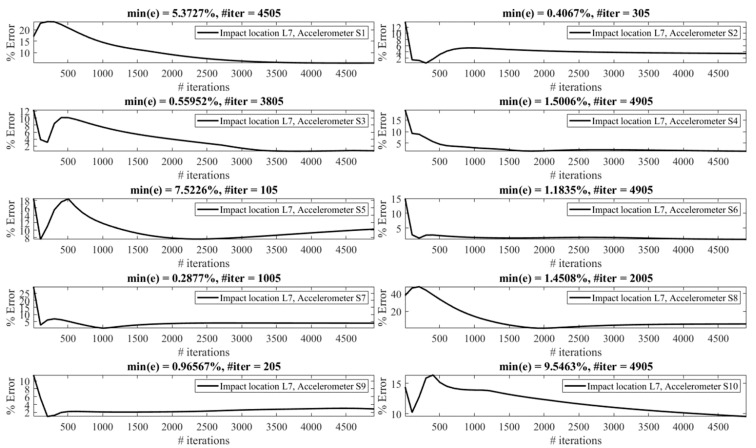
Reconstruction error at impact location L_7_ for different number of iterations in Landweber regularization, using accelerometers placed at S_1_ to S_10_.

**Figure 20 sensors-23-09257-f020:**
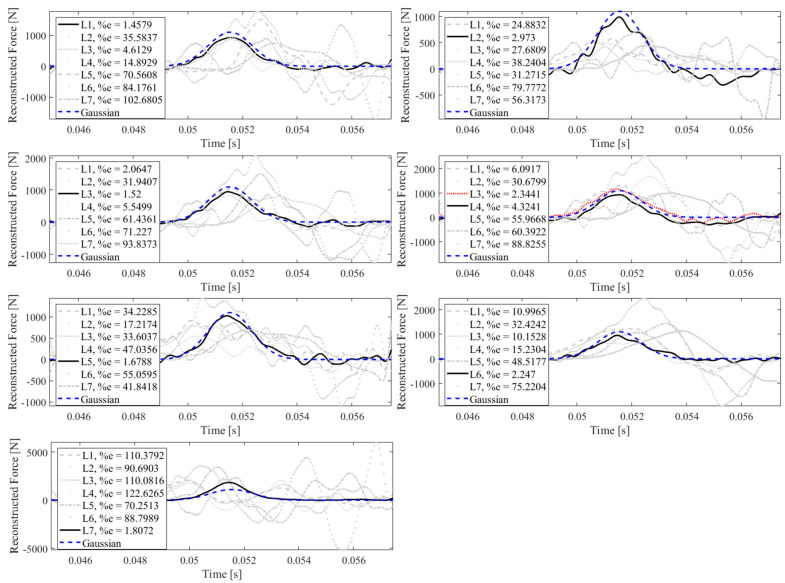
Identification of impact location using accelerometer S_7_ at all impact locations L_1_ to L_7_ with the correlation errors (1-cv)% reported.

**Figure 21 sensors-23-09257-f021:**
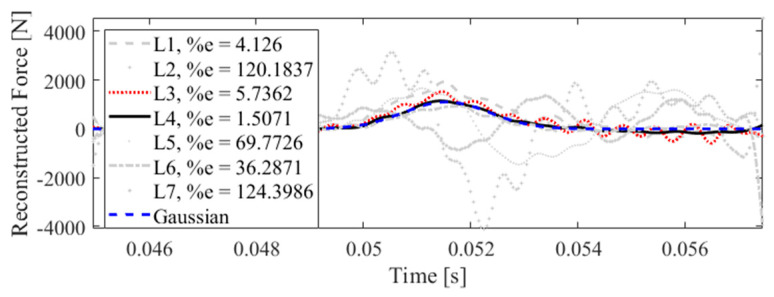
Identification of impact location using accelerometer S_4_ at true impact location L_4_ with the correlation errors (1-cv)% reported.

**Table 1 sensors-23-09257-t001:** The impact reconstruction error for different impact locations, using different accelerometers.

		Sensor Position									
		S_1_	S_2_	S_3_	S_4_	S_5_	S_6_	S_7_	S_8_	S_9_	S_10_
**Impact location**	L_1_	9.04	4.09	14.97	3.36	25.32	35.74	5.32	22.12	16.45	12.10
	L_2_	47.51	10.46	45.85	10.42	10.30	12.69	4.82	51.03	3.82	8.05
	L_3_	11.91	3.96	17.50	11.04	10.60	8.44	4.27	28.47	15.55	7.86
	L_4_	67.59	33.17	50.34	8.34	12.51	27.00	17.78	77.05	26.17	21.70
	L_5_	63.42	4.64	19.68	7.11	8.52	1.37	1.68	17.54	3.64	5.86
	L_6_	11.07	6.42	3.07	5.28	7.27	1.08	4.45	4.06	3.58	19.54
	L_7_	7.33	4.05	2.79	2.10	7.63	1.85	3.94	3.26	2.33	11.62

The green-, yellow-, and red-colored cells, respectively, demonstrate errors less than 5 percent, between 5 to 10 percent, and above 30 percent.

**Table 2 sensors-23-09257-t002:** The impact reconstruction error with an optimal value of the number of iterations for different impact location, using different accelerometers.

		Sensor Position									
		S_1_	S_2_	S_3_	S_4_	S_5_	S_6_	S_7_	S_8_	S_9_	S_10_
**Impact location**	L_1_	9.04	3.64	6.28	2.91	23.95	26.85	3.03	18.19	13.66	10.80
	L_2_	21.24	3.18	13.31	6.13	9.55	4.83	4.01	14.72	3.12	7.54
	L_3_	7.81	3.45	11.31	8.19	10.32	7.94	3.72	10.27	13.02	4.08
	L_4_	30.97	8.00	6.31	3.61	11.20	14.12	9.05	33.32	15.05	21.70
	L_5_	14.86	3.74	8.13	3.95	1.87	0.79	1.66	15.79	3.15	2.01
	L_6_	10.64	5.12	3.07	3.55	7.12	0.84	4.20	4.01	1.64	17.79
	L_7_	5.37	0.41	0.56	1.50	7.52	1.18	0.29	1.45	0.97	9.55

The green-, yellow-, and red-colored cells, respectively, demonstrate errors less than 5 percent, between 5 to 10 percent, and above 30 percent.

## Data Availability

The data presented in this study are available on request from the corresponding author.

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
