# Peer review of "Iterative-Based Impact Force Identification on a Bridge Concrete Deck"

_sensors, 2023, doi:10.3390/s23229257_

Round 1
Reviewer 1 Report
Comments and Suggestions for Authors
This study proposed to reconstruct the impact force by using transfer matrix of the structure and Landweber method, and the location of impact force is automatically determined by comparing the reconstructed impact force with Gaussian profile. Then, the proposed method is applied to lab test model with 7 impact force locations and 10 accelerometers, and the accuracy of the proposed method are discussed. However, this paper still lacks of detailed explanation on the ingredients and implementation of the identification method, as well as its application prospects. The authors should address the following comments on the paper before it can be accepted for publication.
Q1: Some of the equations are not correctly displayed in the manuscript, as well as the typos. The following are some examples to be modified:
Line 101, Equation (2);
Line 157, Equation (7);
Line 234, Table 2, row name is not correctly displayed;
Line 320, since, in this table, an optimal,,,, inconsistent font size;
Besides, the affiliations of authors should be checked again because there are two affiliations both noted as d. The authors should carefully check the whole paper to ensure the layout and readability of the paper.
Q2: In introduction part, line 79 to 83, it is said the main three contributions of this work are: (i) introduction of low-pass filter to improve construction accuracy; (ii) standardized accuracy error metric is utilized to improve the evaluation of the reconstruction validity; (iii) Gaussian profile is used to automatically detect the location of impact force. However, the following details should be explained more in order to clearly illustrate the above contributions.
(1) For the introduction of low-pass filter, it is not clear that why and how the reconstruction accuracy can be improved by filtering the high frequency component of the signal. For example, as illustrated in line 192 to 194, how the parameters of low-pass filter is determined to remove high-frequency noise? Is it done by trial-and-error test or based on the properties of structure and sensor? Also, since low-pass filter remove the high frequency component, in Fig. 11 the subplot of Location L5 shows that the error of filtered reconstruction will increase with the increase of iteration in Landweber Regularization, does this will influence the selection of low-pass filter?
(2) For the introduction of Gaussian profile, the true impact force is the one which has the minimum correlation error with Gaussian profile. However, to the reviewer’s understanding, the Gaussian profile given in Eq. (11) is determined based on the similarity between its shape and the shape of actual impact force, and the effectiveness of this automated location detection is questionable if the impact force fails to have such Gaussian-profile like shape, since it may happens to have such shape only with the experiment settings in this paper.
Q3. To the reviewer’s understanding, the iteration of Landweber Regularization plays an important role in the proposed identification method, and it is determined subjectively based on the error comparison of pre-test results. However, as shown in Figs. 13-19, the iteration which has the minimum error of each subplot vary significantly and some of them even exceeds 2500. So it is confused that how the authors came into a conclusion that 2500 iterations is the optimal value for impact force reconstruction without giving further details, for example the impact reconstruction error with other iteration number can be also listed as Table 3 in the paper for comparison purpose.
Q4. Although the proposed method is a model-based method, and the applicability is also discussed in Sec4.5, the effectiveness of the proposed identification method is still questionable since only the optimal results obtained by sensor 7 is used for final location identification, while there are totally 7 sensors to collect vibration data. Could the authors implement the location identification process with the same Gaussian profile by using data from another reliable sensor? Such that both results could be used to cross-validate the impact force location. Besides, the author should address how to ensure reliable sensors during the measurement in the paper, since it is concluded in Sec.5 that particular sensors can lead to more precise results which can be discovered by pre-made tests.
Q5. In Sec 4.4, line 389, it is concluded that using the closest available sensor to reconstruct an impact force can improve the precision, however, the effective sensors 7 and 9 are not the closer to some of the impact load locations than other sensors, while they both have good accuracy as stated in the paper. The author should explain more to draw such conclusion, for example, for reliable sensor.
Q6. In addition, the authors may want to double-check the formulae, like (2)&(7), there are some typos that may be caused by format conversion.
Q7. The authors mentioned that they use a Gaussian profile to express the overall impact event profile, it is better to explain the reasons or cite papers to support that. Intuitively, in terms of structure response under an impact, it should be asymmetric (a sharp ramp-up followed by an exponential decay). Can you explain that?
Q8. In Formula 11, the authors mentioned that the parameters were chosen arbitrarily. It is confusing here, to match the real case, should not there parameters be derived by using model regression?
Q9. Regarding the number 2500 for iterations, the authors mentioned that substantial changes occur almost before 2500 iterations at lines 276 & 277. What is the exact meaning of substantial change here?
Q10. In Figs. 13 ~ 19, as shown, the error does not simply go lower as the number of iteration increase. Some subplots show small fluctuations while decreasing, and some show sharp fluctuations. In some subplots, the curves even go higher as the number of iterations increases. Is that due to the Landweber regularization used in the algorithm? It would be helpful for the readers to understand the algorithm by providing more detailed explanations of these observations.
Comments on the Quality of English Language
The quality of English Language is okay, but need more polishing.
Author Response
Dear respected reviewer,
Please see attached document.
Best regards,
Hamed

Reviewer 2 Report
Comments and Suggestions for Authors
The paper
“Iterative-Based Impact Force Identification on a Bridge Concrete Deck”,
By
Rashidi et al.,
Presents an investigation on the application of the Landweber method for the identification of impact forces on steel-reinforced concrete decks, which are commonly used in civil structures such as road and railway bridges.
The paper can be of good interest to researchers and practitioners in civil engineering. It is definitely worth attention but it presents some conceptual and editorial issues. Thus, before being fully accepted, the following major and minor remarks should be all fully addressed.
Major (conceptual) remarks:
1. Indeed, bridge decks are prone to unexpected impact forces during their operational lifespan, and precise identification of these impact events is crucial for structural health monitoring. However, it is not clear if the force impressed by the PCB impact hammer (>=200 N?) can be considered a realistic representation of actual, real-life impact forces. Perhaps it is even smaller than many potential damage-inducing impacts, in which case, the methodology could be even more sensitive than the strictly required (and thus on the safe side).
2. Related to the previous comment, it is not clear how (and if) the input forces have been designed for the experimental validation. E.g. for L1 (Figure 4) the peak amplitude is 2000 N, while it is around 500 N for L5 (Figure 8). Some variability is understandable due to the hammer hits, but it is not clear if this large difference (4x) was random or intended to verify different load cases.
3. The experimental case study is intended to replicate the decks of the Sydney Harbour Bridge in Australia. Is there any specific reason for this particular design choice? Is this steel-reinforced concrete structural configuration particularly common (such that the results can be applied to a wide range of other, similar structures) ?
4. The state-of-the-art review can be expanded, especially including recent developments in the Bridge Monitoring of concrete deck road bridges, such as https://doi.org/10.1016/j.engstruct.2022.115573 and similar ones.
5. In general, the results shown in the Figures of Sec 4.1 should be commented on in more detail.
6. Overall, as a general comment, it seems from all figures that the reconstructed force time series is (most of the time) very valid around the peak, but suffers from some ‘surface undulations’ moving far from it. Is this perhaps a limitation due to the mathematical formulation of the Landweber method?
Editorial and minor remarks:
1. The Type of Paper should be indicated in the top left corner.
2. Eq. 2, 7 have some issues (Chinese characters superimposed).
3. Figures 4 -10, 11-12, and 20-21: the legend covers too much of the plot (especially, part of the curves). This is a less severe issue for the other similar charts (Figures 13-19)
4. Again on Figures 4 to 10 (and the following ones): as editorial advice, it would be better to alternate figures and text rather than clump all figures together one just after the other.
5. Eq 11: the meaning of the symbol ‘*’ must be specified. If it only represents simple multiplication, the dot symbol ‘ ’ should be preferred, as the star is also commonly used to indicate convolution.
6. The DOI should be reported for all the References in the Bibliography.
Comments on the Quality of English LanguageThe English of the paper is overall good, yet it may use some grammar checking. There are a few minor mistakes and typos throughout the manuscript (e.g. line 133 “Section 1,,”).
Author Response

(The authors gave the same response as above.)

Round 2
Reviewer 1 Report
Comments and Suggestions for Authors
Thanks for your efforts! Please find the additional comments in the attached file.

Comments on the Quality of English LanguageLooks good.
Reviewer 2 Report
Comments and Suggestions for Authors
This Reviewer is overall satisfied with the changes made by the Authors and the replies to the specific remarks. The article can be accepted after the following minor issues have been addressed:
1. Equations 2,7, and elsewhere, there are still Chinese characters superimposed
2. Table 1, the third row is not clear (200 18?)
3. The issue with the legends of Figures 4 -10, 11-12, and 20-21 has not been completely solved.
4. Please fill in the missing information at the end of the paper: Author Contributions, Funding, Institutional Review Board Statement, Informed Consent Statement, Data Availability Statement, and Conflicts of Interest.
Comments on the Quality of English LanguageThe English of the paper has been improved with respect to the previous draft
